# Lead acetate–based test strip method for rapid and quantitative detection of residual sulfur dioxide in Chinese herbal medicines

**Penghua Zhao**[1], **Xiaoyan Huang**[2], **Yaping Li**[1], **Haixiang Zhang**[1], **Qiyu Wang**[3], **Dongliang Li**[3], **Cuixiang Xu**[1,2]*, **Jianhua Wang**[1,2]

**1** Research Center of Cell Immunological Engineering and Technology of Shaanxi Province, Central Lab of Shaanxi Provincial People's Hospital, Xi'an, China, **2** Shaanxi Provincial Key Laboratory of Infection and Immune Diseases, Shaanxi Provincial People's Hospital, Xi'an, China, **3** Department of Graduate School, Yan'an University, Yan'an, China

* xucuixiang1129@163.com

**Data Availability Statement:** All relevant data are within the manuscript and its Supporting Information files.

## Abstract

To prevent the residual sulfur dioxide in Chinese herbal medicines (CHM) caused by sulfur fumigation, which may lead to severe health issues, there is an urgent need for a rapid and quantitative detection technique. Sodium borohydride was used as a reducing agent to convert sulfur dioxide into hydrogen sulfide, which was then detected using lead acetate test strip. An accurate testing apparatus was designed, consisting of reaction bottle cap, reaction bottle, lead acetate test strip, and sulfur dioxide detector. The effect of different reaction variables on detection, including reductant quality, pH of initial media, reaction time, lead acetate concentration, and membrane type was investigated. The optimal conditions were determined by orthogonal experiments. The reaction membrane type and lead acetate concentration on the membrane were optimized to enhance detection accuracy. Standardized gray cards were fabricated and used to calibrate the detector. The detection system demonstrated an exceptional linear correlation (r2 = 0.9992), with a linear detection range of 0–750 mg·kg$^{-1}$. The colored substances and sulfur-containing substances within the matrix of CHM did not affect the detection results. Therefore, the detection method exhibited superior accuracy and stability. The proposed technique proved to be swift, reliable, and provides a straightforward and convenient approach for the quantitative determination of sulfur dioxide in CHMs. The results of this work may provide insights into the development of test strips for quantitative detection.

## 1. Introduction

Sulfur fumigation is a traditional method used for processing CHM, in which sulfur dioxide (all the different forms of sulfur dioxide in equilibrium, including molecular sulfur dioxide, bisulfite, and sulfite, will be collectively referred to as "SO$_2$" throughout this text) is generated on the surface of the herb through the combustion of sulfur. This process not only imparts vibrant color to Chinese herbal medicines (CHM) but also eliminates bacterial and insect

**Funding:** This work was supported by the Key Research and development program of Shaanxi Province(2024SF-YBXM-081), Shaanxi provincial People's Hospital Science and Technology Personnel Support Program(No.2021LJ-02), Key Research and development program of Shaanxi Province(No.2022KWZ-20), Shaanxi provincial People's Hospital Science and Technology Development Incubation Fund (2023YJ-22) Shaanxi Provincial health and Medical Research Innovation Capacity Enhancement Plan(2024PT-01), Shaanxi Province Innovation Capability Support Program Project (S2024-ZC-TD-0127), Shaanxi Provincial health high-level talent cultivation program, the Shaanxi Special Support Plan for High Level Talents. The funders had no role in study design, data collection and analysis, decision to publish, or preparation of the manuscript.

**Competing interests:** The authors have declared that no competing interests exist.

contaminants in these materials. However, excessive levels of $SO_2$ residue can have detrimental effects, as they have the potential to not only alter the properties and chemical composition of CHM [1,2] but also compromise their quality by damaging bioactive compounds [3], leading to various degrees of harm for users [4,5]. Research findings indicate that exceeding $SO_2$ residue limits is a common occurrence in CHM [6]. However, as the storage and usage of CHM are deeply rooted in traditional processing methods, a complete ban on sulfur fumigation is not feasible [7]. In order to prevent the abuse of sulfur fumigation, the Chinese pharmacopoeia commission stipulated the limit of sulfur dioxide as early as the second supplement of the 2010 edition of the Chinese Pharmacopoeia of the People's Republic of China. These limits vary depending on the type of CHM, with two thresholds set at 150 mg·kg$^{-1}$ and 400 mg·kg$^{-1}$ [8]. In 2020 edition of Chinese Pharmacopoeia, excessive sulfur fumigation is still an important problem affecting the quality of Chinese medicinal materials.

Commonly employed techniques for detecting $SO_2$ in CHM include acid-base titration [9], ion chromatography [10], gas chromatography [11], fluorescence derivatization [12], and colorimetry [13]. And the SERS [14] and quantum dot methods [15] developed in recent years. The titration iodometry which is based on acid-base neutralization is simple and convenient, but for samples with high organic acid content, it is difficult to determine the end point of titration, which will lead to inaccurate method [16] and extended time requirements [17]. The results obtained by Gas chromatography and ion chromatography are accurate, but the specificity is poor, the detection efficiency is low, and the operating conditions are demanding [18] and need sophisticated detection equipment [19]. Whereas fluorescence derivatization has poor accuracy and reproducibility, a narrow linear range [20], and high cost [21]. The colorimetry is polluting the environment [22] and can not be compared with colored Chinese medicine. Consequently, the rapid and precise detection of residual $SO_2$ in traditional Chinese medicine remains a technical challenge demanding resolution [23].

As is well-known, hydrogen sulfide can be readily detected using a lead acetate test strip. Moreover, under suitable reductive conditions, $SO_2$ (S(IV)) can be transformed into hydrogen sulfide (S(−II)) [24]. By combining these two reactions, it becomes possible to detect $SO_2$ using a lead acetate test strip. In this work, we devised a swift and precise method for quantifying $SO_2$ in CHM by adopting these chemical reactions. The approach was proven to be highly effective, boasting advantages such as ease of operation, remarkable sensitivity, and the ability to circumvent background interference from CHM. The chemical equation is as follows:

$$SO_2 + 2OH^- \rightarrow SO_3{}^{2-} + H_2O$$
$$SO_3{}^{2-} + BH_4{}^- + 4H^+ \rightarrow H_2S \uparrow + BO_2{}^- + H_2O \qquad (1)$$
$$H_2S + Pb^{2+} + 2CH_3COO^- \rightarrow PbS \downarrow + 2CH_3COOH$$

## 2. Materials and methods

### 2.1. Reagents, materials, and apparatus

Lead acetate, sodium borohydride, sodium sulfite acid, ferrous sulfide, sodium hydroxide acid, and hydrochloric acid (all of analytical purity) were procured from Sinopharm Chemical Reagent Co., Ltd. (Shaanxi, China, https://www.sinoreagent.com/). Nitrocellulose membranes (Millipore M135, Millipore M180, Sartorius CN95, Sartorius CN140, MDI E03 3Mtr, MDI E21 90Mtr) were obtained from Kinbio Tech Co. (Shanghai, China).

A $SO_2$ reaction bottle cap was obtained from Shanxi Ruiqi Biotechnology Co., Ltd. (patent no: Zl201620703797.6). An RQS-100 $SO_2$ quantitative detector, a handheld $SO_2$ detector, was

acquired from Shaanxi Ruiqi Biotechnology Co., Ltd. (Patent No. 2019SR0608587), utilizing system software V1.0.

An XPE105 electronic analytical balance was acquired from Sartorius AG (Gottingen, Germany). An SHZ-B thermostatic oscillator was purchased from Shanghai Boxun Co., Ltd. (Shanghai, China). A CNC induction chopper was obtained from Shanghai Jinbiao Biotechnology Co., Ltd. (Shanghai, China). Double-distilled water was generated using a Millipore Milli-Q water system (Bedford, MA, USA).

## 2.2. Methods

**2.2.1. Design of the detection device.**　To ensure the utmost precision of the detection results, we designed an apparatus for detecting $SO_2$, comprising a reaction bottle cap, a reaction bottle, a lead acetate strip, and a detector. The 25-mL reaction bottle was made of thickened boric acid glass and had a bottle-mouth inner diameter of 12 mm. The reaction bottle cap was made of silicone and could form a secure seal with the reaction bottle, featuring a 5-mm-diameter hole in the center. A silica gel guide groove was situated on the bottle cap to ensure the accurate coverage of the center hole by the lead acetate film on the test strip. The lead acetate strip was a 7 mm × 52 mm polyvinyl chloride strip featuring a 5-mm-diameter hole, which was used to apply the lead acetate reaction membrane. The detector comprised three main components: a light source system, an image acquisition system, and an image analysis system. The light source employed central ring circuit illumination with high-intensity white light-emitting diodes, ensuring an illumination uniformity across the entire field of view of no more than 2%. The image acquisition system employed a charge-coupled device digital camera, specifically a 1/3" SSE2514 with a 25-mm F1.4 Avenir CCTV lens. During the scanning of the grayscale card, the highly photosensitive semiconductor material converts light into electrical charges, which are subsequently transformed into digital signal voltages through an analog-to-digital converter chip. This converted digital signal is then passed to the image acquisition card to be transformed into digital image signals, which are further processed by the image analysis system to calculate the grayscale integral value. Through appropriate conversion, this value can ascertain the content of $SO_2$ in CHM. Fig 1 provides a visual representation of the experimental apparatus.

**2.2.2. Experimental procedure.**　The detection of $SO_2$ in CHM was carried out in reaction bottles. First, a 0.4-g sample of CHM was added to the reaction bottle. Then, 2 mL of deionized water was added to the bottle, with its initial pH ranging from 5 to 13 (adjusting by 1 mol/L HCl or 1 mol/L NaOH). Next, the mixture was mixed thoroughly by shaking at 100 rpm for 10 min at room temperature before the reaction. A specially prepared lead acetate test strip was then inserted into the reaction bottle cap, and the lead acetate detection membrane was saturated with 7 μL of double-distilled water. Then, an appropriate amount of $NaBH_4$ was added to the reaction bottle, which was immediately sealed with the prepared reaction bottle cap. Then 2 mL of 2 mol/L HCl was added to the reaction bottle by using a syringe by piercing the silicone reaction bottle cap. Finally, the mixture was reacted thoroughly by shaking at 100 rpm for 1–30 min at room temperature. The reaction bottle was allowed to react for 6 min until the color stabilized. The detection instrument was then employed for analysis.

**2.2.3. Optimization condition of chemical reaction.**　Orthogonal design provides a simple and systematic approach for design optimization of quality, performance, and cost. The selection of control factors is the most important stage in the experimental design, as it is possible to identify non-significant variables and relatively important variables at early stages by including as many factors as possible [25]. To reduce experimental time and research costs, three fundamental factors were considered in a three-level three-factor orthogonal array (L9)

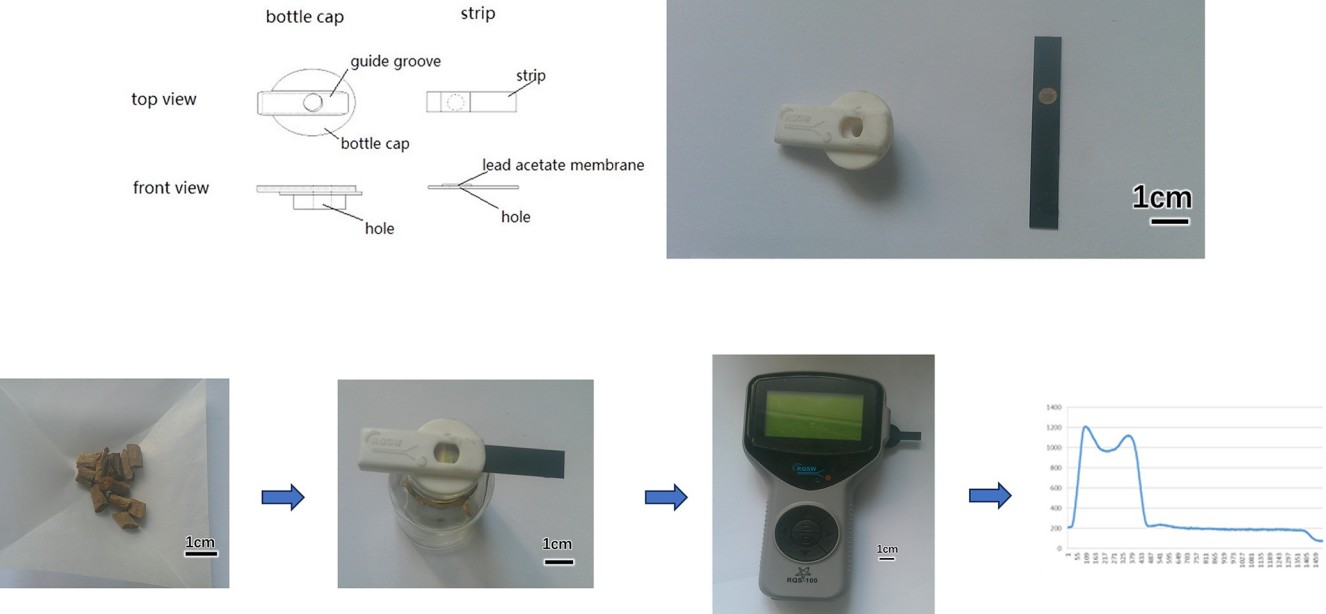

**Fig 1. Schematic illustration of lead acetate-based strip detection of residual sulfur dioxide in Chinese herbal medicines.** a: Schematic of the basic structure of the bottle cap and the lead acetate test strip (the figure is similar but not identical to the original image and is therefore for illustrative purposes only). b: Diagram of the process of sulfur dioxide detection.

to determine the optimal (or most cost-effective) level affecting the reaction rate: $NaBH_4$ concentration, reaction time, and initial pH of solution (i.e., acidic, neutral, or basic) (Table 1).

**2.2.4. Optimization of test strip.** Choosing the correct membrane for use in the lead acetate–based strip method is crucial for the accurate detection of $SO_2$ in CHM. It is widely recognized that although nitrocellulose (NC), nylon, and polyvinylidene fluoride (PVDF) membranes can all be used as strip membranes, nylon has unstable adsorption properties, and PVDF membranes require pre-wetting with methanol for use. Therefore, we have chosen NC membranes as the material for the membrane. It is important to note that lead acetate will be adsorbed to the polymer matrix of the NC membrane, but not in the membrane gap, and the signal of the membrane surface is only observable to the human eye at a depth of less than 10 μm. Thus, the amount of lead acetate adsorbed on the membrane can affect the apparent color development, and the effect of lead acetate concentration on the membranes should be considered. Different manufacturers, and different types of NC film, will produce different color effects, so we should test the effect of different types of membranes from different manufacturers on the accuracy of the test.

*Effect of different concentrations of lead acetate on detection.* The Sartorius CN95 membrane served as the initial substrate for the lead acetate utilized in the $SO_2$ detection strips. To explore the impact of lead acetate concentration on detection accuracy, lead acetate solutions with

**Table 1. Experimental factors and levels selected for the sodium borohydride reduces sulfur dioxide to hydrogen sulfide.**

| Experimental factors | Parameters | Level 1 | Level 2 | Level 3 |
|---|---|---|---|---|
| $F_a$ | NaBH4 concentration (mM) | 0.5 | 5.0 | 50.0 |
| $F_b$ | Time (min) | 1.0 | 10.0 | 30.0 |
| $F_c$ | Initial pH | 5.0 | 7.0 | 13.0 |

concentrations of 5%, 10%, 15%, and 20% were prepared to produce test strips of varying concentrations. The membranes were fully immersed in the lead acetate solutions and subsequently dried at 37°C for 4 h.

The detection substance employed in this study was an $SO_2$ standard of gradient concentration, with each gray value of spots obtained from the reaction measured and recorded. By charting the relationship between the gray value of the spots and the concentration of $SO_2$, the effect of different lead acetate concentrations on detection accuracy was analyzed.

*Effect of different membrane types on detection.* It is important to note that differences in the manufacturing process or material composition of different brands of nitrocellulose membrane can have an impact on color presentation. We fabricated test strips by employing NC membranes of varying types from diverse manufacturers. These membranes were impregnated with a 15% solution of lead acetate. Subsequently, we utilized gradient concentrations of $SO_2$ as the target analyte and plotted the correlation between the gray values of the resulting spots and the $SO_2$ concentrations. This allowed us to assess the impact of the varying membrane materials on the precision of detection.

**2.2.5. Preparation of standards.** *Preparation of sodium sulfite standard solution.* After sulfur fumigation, sulfite is the main form in which sulfur exists in CHM. Therefore, a sodium sulfite ($N_{a2}SO_3$) standard solution was prepared to simulate sulfur-fumigated CHM. To prepare the solution, 1 g of $N_{a2}SO_3$ was accurately weighed and added to a 100-mL volumetric flask. Distilled water was then added to the flask to scale, resulting in a 10 g·kg$^{-1}$ solution. From this solution, volumes of 0.2 mL, 0.6 mL, 1 mL, 1.6 mL, 3 mL, 6 mL, 8 mL, 10 mL, 12 mL, and 16 mL were measured and placed into separate 100-mL volumetric flasks. Double-distilled water was then added to each flask to scale, resulting in solutions with concentrations of 20 mg·kg$^{-1}$, 60 mg·kg$^{-1}$, 100 mg·kg$^{-1}$, 160 mg·kg$^{-1}$, 300 mg·kg$^{-1}$, 600 mg·kg$^{-1}$, 800 mg·kg$^{-1}$, 1000 mg·kg$^{-1}$, 1200 mg·kg$^{-1}$, and 1600 mg·kg$^{-1}$; these solutions corresponded to $SO_2$ concentrations of 10 mg·kg$^{-1}$, 30 mg·kg$^{-1}$, 50 mg·kg$^{-1}$, 80 mg·kg$^{-1}$, 150 mg·kg$^{-1}$, 300 mg·kg$^{-1}$, 400 mg·kg$^{-1}$, 500 mg·kg$^{-1}$, 600 mg·kg$^{-1}$, and 800 mg·kg$^{-1}$, respectively.

*Preparation of sulfur dioxide quality control samples.* To account for potential variations between the determination results of standard solutions and those of actual samples, and to better reflect the true state of the samples in question, a quality control procedure was adopted in this study. Specifically, quality control samples were prepared using actual samples that were simulated by diluting the $SO_2$ standard with an extraction matrix mixture of CHM. All CHM used in this study were previously tested using the ion chromatography method mandated by the government and confirmed to not contain any $SO_2$. This approach allowed for a more accurate and representative assessment of the impact of the CHM matrix on the detection results.

**2.2.6. Sulfur dioxide detection accuracy in Chinese herbal medicines.** *Grayscale gradient detection cards for quantitative detection of sulfur dioxide.* The detection principle used in this work is based on the formation of black lead sulfide precipitate through a reaction with $SO_2$. Detection is achieved by measuring the gray value of the precipitate. The ability of the detector to recognize and identify grayscale spots is crucial, which is why a series of grayscale gradient detection cards were designed to ensure effective recognition. To meet the requirements of the Pharmacopoeia of the People's Republic of China for $SO_2$, 10 grayscale gradient strips were designed. The debugging program of the $SO_2$ quantitative detector was used to detect the waveform of the standard test strips.

*Investigating the linear relationship between gray value and standard solution concentration.* The quality control samples for $SO_2$ were detected using the optimal conditions. The samples underwent three parallel experiments, producing gray spots that were then measured using the $SO_2$ quantitative detector to obtain reading values. The linear relationship between the average

gray level value and the concentration of standard quality control solution was investigated using a standard curve.

*Interference experiment*. The above-mentioned detection method relies on the color reaction of the test strip. However, the matrix pigments of CHM can potentially interfere with the detection results. To examine the impact of these matrix solutions on detection, samples were prepared using matrix solutions of CHM(*Lycium barbarum*, *saffron*, and *rhubarb*) known for their propensity for color interference.

Extracts of three CHM(*Lyceum barbarum*, *saffron*, *rhubarb*), which were confirmed to be without $SO_2$, were used to dilute the $SO_2$ standard to create test samples. Water was also used to dilute the $SO_2$ standard to create a parallel control. The effect of matrix color from the CHM and sulfur-containing substances within the CHM matrix on the detection accuracy was then investigated.

*Repeatability testing*. The stability of a method is measured by its repeatability, which refers to the variation in results obtained from multiple measurements of the same sample under the same conditions. The stability of the $SO_2$ detection method used in this work was assessed by determining the repeatability at various sample levels according to the method of Section 2.2.2. The deviation of repeated measurements of the same sample using this method was calculated, and the detection stability was evaluated.

*Stability testing*. To ensure the scientific soundness and potential for future development of the proposed detection method, it was essential to evaluate its stability. In this study, the reagents employed in the detection method were stored in a refrigerator at 2–8˚C. Unopened reagents were collected at 0, 2, 4, 6, 8, 10, and 12 months from the date of storage, and were subjected to testing in accordance with the test method. About the detection substance and its concentration, we consulted the national standard detection line; designed high, medium, and low concentration gradients; selected three different CHM extracts as matrices; and conducted three parallel measurements for each sample to investigate the storage stability of the method.

**2.2.7 Application verification.** 10 samples of different batches of Chinese medicinal materials were collected, 0.4 g, respectively, according to method 2.2.2, and tested under optimal conditions. Each sample was determined three times, and the average value was compared as the final value. At the same time, according to the third method of Chinese Pharmacopoeia (2015 edition), each sample was determined three times and the average value was compared as the final value. The results of the two methods are compared, so as to achieve the application verification of the detection system.

# 3. Results and discussion

## 3.1. Optimization of various conditions of the detection system

### 3.1.1. Optimization of reduction conditions by orthogonal array experiments.
Tables 1–3 show the design and results of the orthogonal array used for the reductive sulfurization process. For each factor in row $V_i$, the results of the three experiments consisting of level *i* were added and then divided by 3, which gave the mean values of $V_i$. For example, the value of $V_2$ for factor $F_a$ is (86.6+100+100)/3 = 95.5. A higher mean value indicated a better sulfur reducing capacity. The range value for each factor was obtained by subtracting the minimum value from the corresponding maximum value among the $V_i$ rows.

Other conditions:100rpm of shaking rate, room temperature, 15% Lead acetate and Sartorius CN95 membrane.

Table 3 shows that the range value of factor $F_b$ was the highest among the three factors, implying that reaction time had a significant effect on reduction of hydrogen sulfide. When the reaction time was increased from 10 min to 30min, sulfide recovery increased from 78.8%

**Table 2. Design and results of experiment for detection processes according to L9(33) orthogonal array.**

| Run No. | $F_a$ $NaBH_4$ concentration (mM) | $F_b$ Time (min) | $F_c$ Initial pH | Recvery of sulfide (%) |
|---|---|---|---|---|
| 1 | 1 | 1 | 1 | 73.3 |
| 2 | 1 | 2 | 2 | 86.6 |
| 3 | 1 | 3 | 3 | 86.6 |
| 4 | 2 | 1 | 2 | 86.6 |
| 5 | 2 | 2 | 3 | 100 |
| 6 | 2 | 3 | 1 | 100 |
| 7 | 3 | 1 | 3 | 76.6 |
| 8 | 3 | 2 | 1 | 90 |
| 9 | 3 | 3 | 2 | 90 |

to 92.2%. Sulfide recovery remained relatively constant at longer reaction times, which was most likely due to the reaction gradually reaching completion. The range value of factor $F_a$ was also high, implying the concentration of $NaBH_4$ had significant influences. Under a $NaBH_4$ concentration of 5 mM, sulfide recovery reached 95.5% (Table 3). Further increasing the $NaBH_4$ concentration from 5mM to 50mM resulted in a decrease in sulfide recovery from 95.5% to 85.3%, respectively. This decrease may have been due to the fact that with the increase of $NaBH_4$ concentration, its reducibility increases; thus, the carbonyl group in the matrix is reduced. Hydrogen sulfide combines with the newly formed carbonyl compounds, resulting in less lead sulfide production [26]. The initial pH ($F_c$) had the lowest effect on the $SO_2$ reduction. Therefore, controlling the initial pH may not be necessary for enhanced levels of $SO_2$ reduction. Based on these results, the optimal reaction conditions were determined to be a $NaBH_4$ concentration of 5 mM and a reaction time of 10 min, with no need to adjust the initial pH value. These results were utilized in subsequent experiments.

**3.1.2. Optimization of test strips.** *Effect of different concentrations of lead acetate on detection accuracy.* The effect of using different concentrations of lead acetate on detection accuracy was investigated using the Sartorius CN95 membrane under a reaction time of 10 min. A series of lead acetate aqueous solutions (5%, 10%, 15%, and 20%) were used to prepare strips, which were employed to detect the concentration gradient of the standard $SO_2$ solution. A plot of $SO_2$ concentration *vs.* detected gray values is presented in Fig 2A. Under a lead acetate concentration of 5%, the detected gray value remained nearly constant as the $SO_2$ concentration increased to 150 mg·kg$^{-1}$. Similarly, under a lead acetate concentration of 10%, the gray value remained stable as the $SO_2$ concentration increased to 400 mg·kg$^{-1}$. At this point, the gray value reached a plateau, indicating that the lead acetate on the membrane reached saturation. In contrast, under lead acetate concentrations of 15% and 20%, the detected gray value increased linearly with $SO_2$ concentration.

**Table 3. Date handling of the L9(33) orthogonal array.**

| $V_i$ | $F_a$ $NaBH_4$ concentration (mM) | $F_b$ Reaction time (min) | $F_c$ Initial pH |
|---|---|---|---|
| $V_1$ | 82.2% | 78.8% | 87.8% |
| $V_2$ | 95.5% | 92.2% | 87.7% |
| $V_3$ | 85.3% | 92.2% | 87.7% |
| Range | 13.3% | 13.4% | 0.1% |

Note: $V_i$, summation of the experimental value of the same level and then divided by 3; Range = max($V_i$)− min($V_i$).

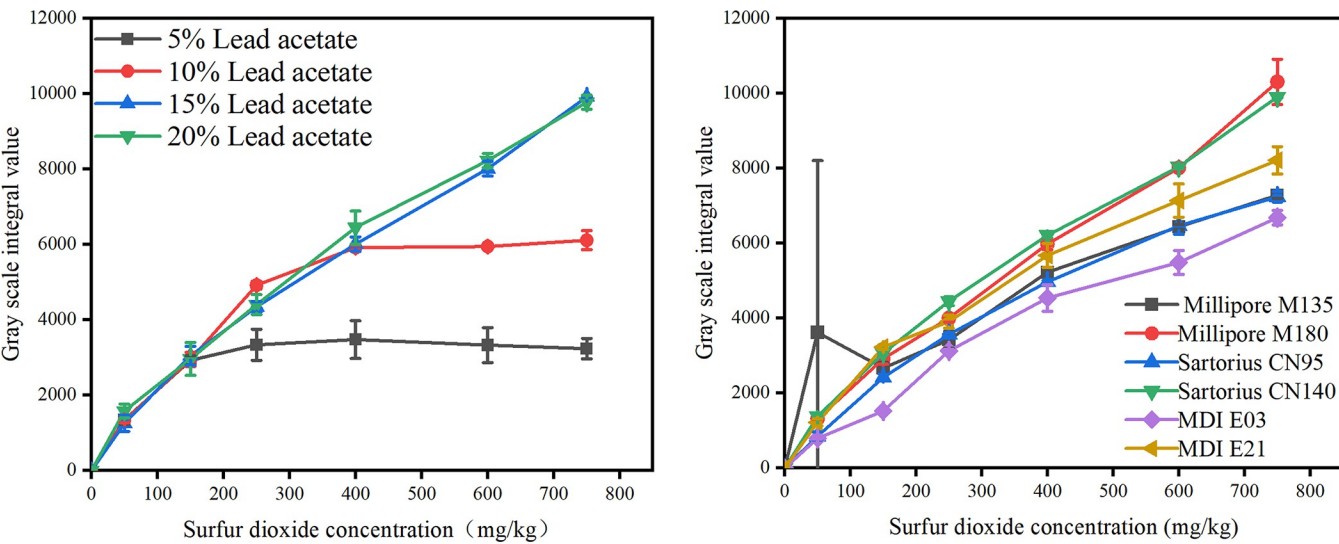

**Fig 2. The relation curve between the gray level and the concentration of sulfur dioxide at different reaction conditions.** a: At different lead acetate concentrations. b: Different types of cellulose membranes.

*Effect of membrane type on detection accuracy*. The effect of different membrane types on detection was investigated under the optimal conditions (lead acetate concentration on NC membrane = 15%; reaction time = 10 min). Test strips with NC membranes from various manufacturers were employed to detect the concentration gradient of the standard $SO_2$ solution. A plot of detected gray value vs. $SO_2$ concentration is presented in Fig 2. Under the optimal experimental conditions, the gray values of films with different types of membrane materials increased linearly with $SO_2$ concentration. The sartorius CN 140 membrane exhibited the best linearity, with an $R^2$ value of 0.9977. Therefore, the sartorius CN 140 membrane was selected as the detection membrane.

## 3.2. Preparation of standard grayscale cards

According to the specific requirements for $SO_2$ in the Pharmacopoeia of the People's Republic of China and the loading capacity of lead sulfide on the NC membrane, test strips with a lead acetate concentration of 15% were prepared. The test strips were inserted into the caps of 11 reaction bottles, and 7 μL of distilled water was used to wet the lead acetate detection membrane placed in each cap aperture. In each of the 11 reaction bottles, 0.5 g (5.68 mmol) of ferrous sulfide was added, followed by the addition of varying volumes (0 mL, 1 mL, 2 mL, 3 mL, 4 mL, 5 mL, 6 mL, 7 mL, 8 mL, 9 mL, and 10 mL) of 2 mol·L$^{-1}$ HCl; the corresponding lead sulfide precipitates were 0 μmol, 0.2 μmol, 0.4 μmol, 0.6 μmol, 0.8 μmol, 1.0 μmol, 1.2 μmol, 1.4 μmol, 1.6 μmol, 1.8 μmol, and 2 μmol, respectively. The reaction bottles were immediately sealed with the bottle caps (with the pre-inserted wetted test strips) and incubated for 30 min to obtain distinct lead sulfide spots of varying intensities. These 11 spots were sequentially denoted as Card 0, Card 1, Card 2, and so on, forming the standard grayscale cards of the detection system. The standard cards are shown in Fig 3. The waveform of each card was detected using the calibration program of the quantitative detector. The waveforms showed a gradient increase in the value of the detected signal for the standard test strips with different gray levels. The electrical signals of the detector for the 11 standard grayscale strips are shown in Fig 3.

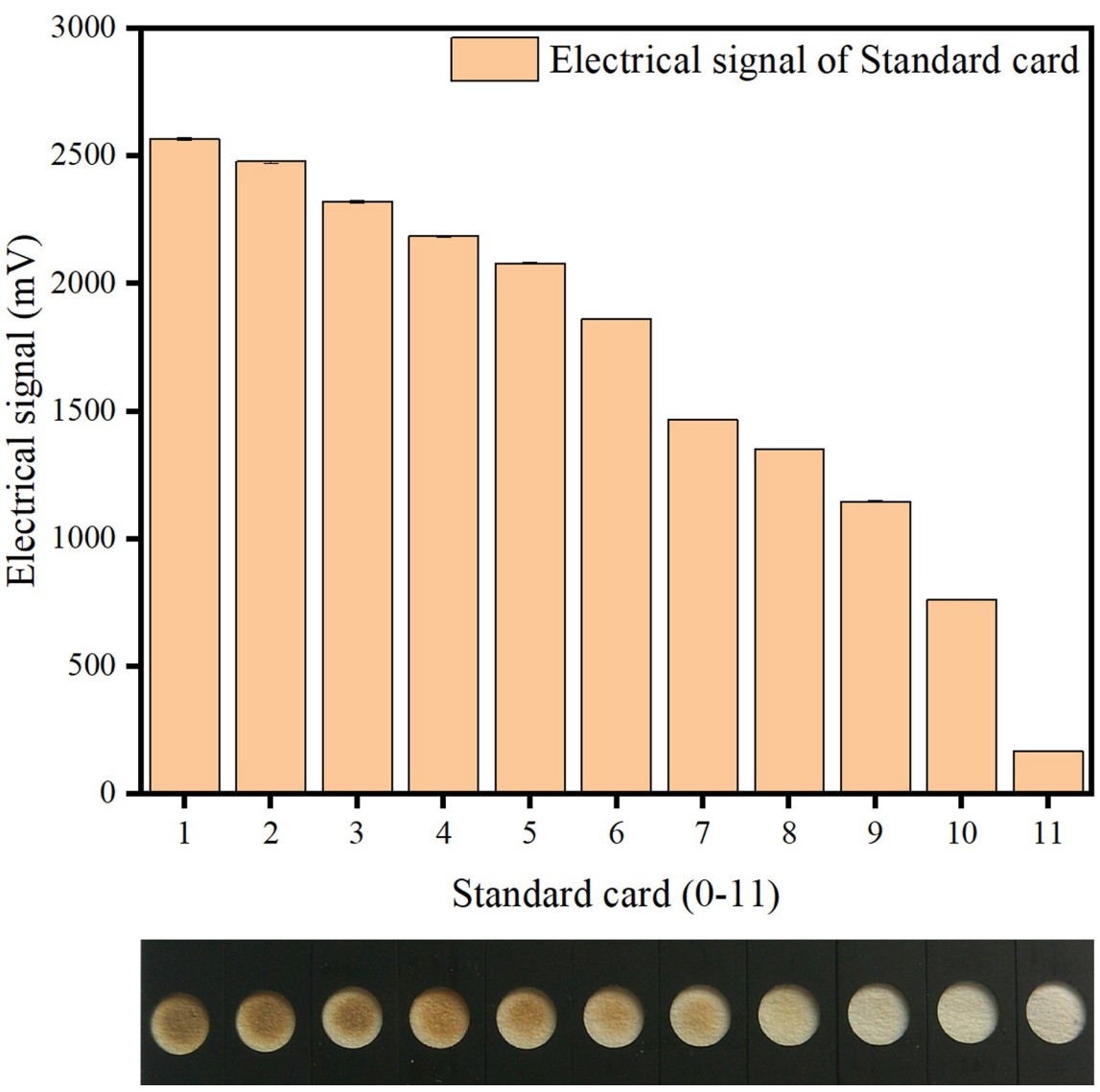

**Fig 3. The 11 standard gray-scale strips and its corresponding elictrical signal.** a: The 11 standard gray-scale strips. b: The corresponding elictrical signal of the 11 standard gray-scale strips.

### 3.3. Standardized sulfur dioxide detection curve for product quality control

After preparing sodium sulfite control solutions with concentrations of 100 mg·kg$^{-1}$, 300 mg·kg$^{-1}$, 500 mg·kg$^{-1}$, 800 mg·kg$^{-1}$, 1200 mg·kg$^{-1}$, and 1500 mg·kg$^{-1}$, corresponding SO$_2$ concentrations of 50 mg·kg$^{-1}$, 150 mg·kg$^{-1}$, 250 mg·kg$^{-1}$, 400 mg·kg$^{-1}$, 600 mg·kg$^{-1}$, and 750 mg·kg$^{-1}$, respectively, were obtained. Following the method described in Section 2.2.2, a series of strips with different gray levels were prepared and tested three times with the SO$_2$ quantitative detector. The average gray value was calculated for each strip, and a standard curve was obtained by plotting the concentration of the SO$_2$ solution against the gray values, as shown in Fig 4. The method showed excellent performance, with an R$^2$ value of 0.9992 and a linear regression analysis of the gray area integral value within the SO$_2$ concentration range of 0–750 mg·kg$^{-1}$.

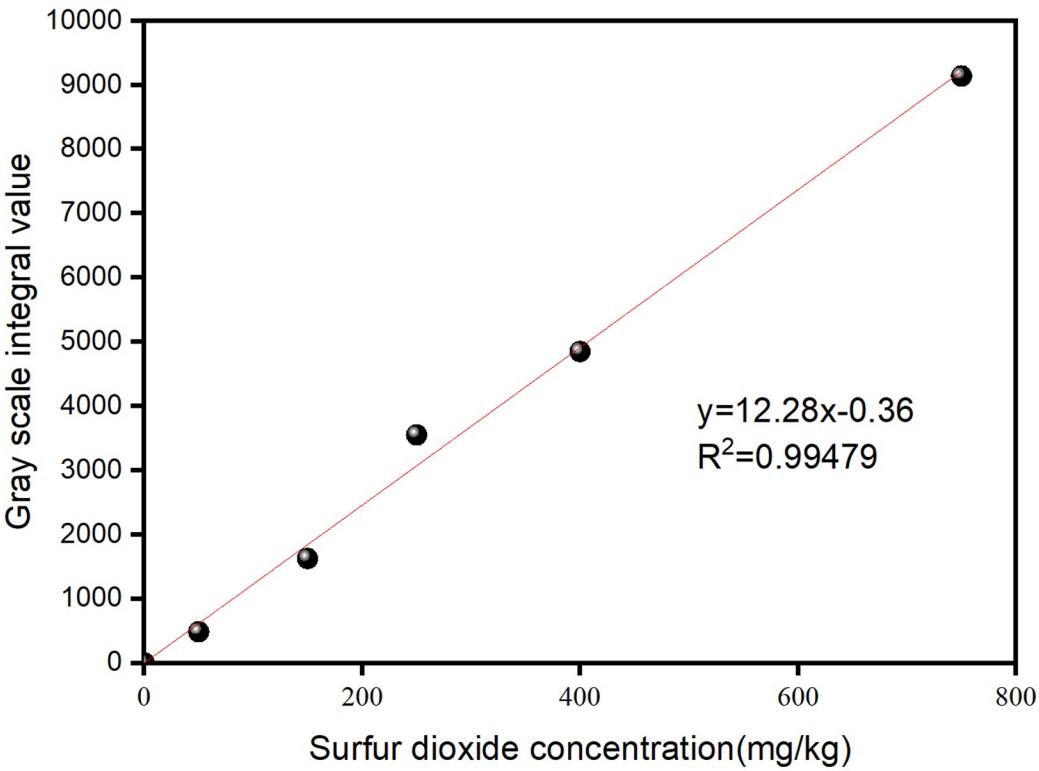

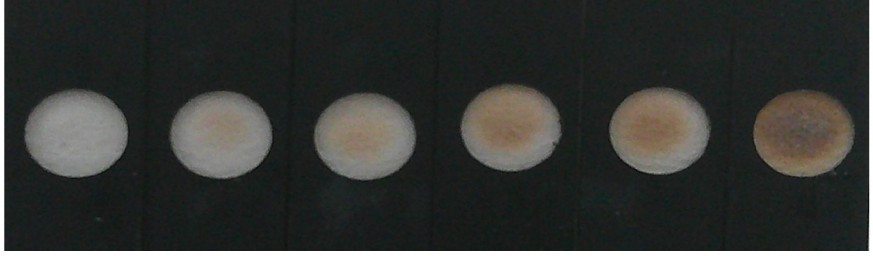

**Fig 4. The standard curve of sulfur dioxide concentration and gray scale.** a: The strip of different concentrations of sulfur dioxide. b: The relationship between sulfur dioxide concentration and gray scale.

### 3.4. Reliability of test system

**3.4.1. Chinese herbal medicine extracts.** First, 0.4 g of CHM (*Lycium barbarum*, *saffron*, and *rhubarb*), confirmed to be free of $SO_2$, was weighed and placed into a sample extraction bottle before adding 2 mL of distilled water. The sample was then soaked and extracted for approximately 5 min by shaking. Then the extracted solution was used to dilute the SO2 standard to prepare the samples with high, medium and low concentrations. The gray value of each sample was detected using the method described in Section 2.2.2, which was converted into concentration using the standard curve shown in Fig 4. The concentration value corresponded with the actual value of the sample, indicating that the CHM matrix did not affect the detection results. This result was likely due to the fact that the detection method is based on the sulfur is changed from solid phase to liquid phase ($SO_2$-$Na_2SO_3$), then to gas phase ($Na_2SO_3$-$H_2S$) and then to solid phase ($H_2S$-PbS) during the detection process, which the colored interfering substances can not enter the next phase. The sulfur-containing substances in

**Table 4. Effect of colored CHM matrix on detection.**

| Substrate of CHM | Level | Actual concentration of SO$_2$ /(mg•kg$^{-1}$) | Detected concentrationof SO$_2$ /(mg•kg$^{-1}$) |
|---|---|---|---|
| *Lycium Barbarum* | High | 400.0 | 400.5 |
| | Medium | 150.0 | 149.5 |
| | Low | 0.0 | 0 |
| *Saffron* | High | 400.0 | 399.7 |
| | Medium | 150.0 | 151.5 |
| | Low | 0.0 | 0 |
| *Rhubarb* | High | 400.0 | 401.9 |
| | Medium | 150.0 | 151.4 |
| | Low | 0.0 | 0 |

CHM also did not affect the detection results. It may be due to the participation of endogenous sulfur-containing substances in hydrogen bonding within the molecule, which affects their reduction by N$_a$BH$_4$ [27]. Another possibility is that the endogenous sulfur compounds have different electron densities on their sulfur atoms, and different steric hindrance, compared with SO$_2$, which affects their reduction by N$_a$BH$_4$ [28]. The results are presented in Table 4.

### 3.4.2. Stability testing

*Precision testing.* Three varieties of CHM, namely, *Pueraria lobata*, *Lycium barbarum*, and *Codonopsis pilosula*, were subjected to SO$_2$ residual concentration detection ten times in parallel for each level. The SO$_2$ residual concentration for the three herbs was was detected by ion chromatography. The three concentrations of *Pueraria lobata* used in precision experiments are 45mg·kg$^{-1}$, 113 mg·kg$^{-1}$ and 251 mg·kg$^{-1}$. The three concentrations of *Lycium barbarum* used in precision experiments are 81mg·kg$^{-1}$,137 mg·kg$^{-1}$ and 205 mg·kg$^{-1}$. The three concentrations of *Codonopsis pilosula* used in precision experiments are 87mg·kg$^{-1}$, 267 mg·kg$^{-1}$ and 664 mg·kg$^{-1}$. The precision of the detection system was evaluated by analyzing the standard deviation and coefficient of variation of the detection results. The test outcomes revealed a high degree of consistency, as evidenced by standard deviation of less than 1.5 and a coefficient of variation of less than 2%. These results are presented in Table 5.

*Preservation stability experiment.* Preservation stability trials were conducted to confirm the method's suitability across diverse CHMs, in which three parallel CHM samples were spiked with SO$_2$ and assayed. Reagents for the detection process were stored in a refrigerator at 2–8˚C. Unopened reagents were retrieved at 0 months, 2 months, 4 months, 6 months, 8 months, 10 months, and 12 months post-storage initiation. Sodium sulfite was incorporated into the extraction reagent of three distinct CHMs—*Lycium barbarum*, *Codonopsis pilosula*, and *Pueraria lobata*—to reach concentrations of 0 mg·kg$^{-1}$, 300 mg·kg$^{-1}$, and 800 mg·kg$^{-1}$, corresponding to SO$_2$ concentrations of 0 mg·kg$^{-1}$, 150 mg·kg$^{-1}$, and 400 mg·kg$^{-1}$, respectively. Samples were subjected to three replicate analyses using the method described in Section 2.2.2, with the mean gray value computed for each. The findings revealed that the gray

**Table 5. The precision of the test result of three kinds of CHM.**

| No. | Name | Repetitive results | | | | | | | | |
|---|---|---|---|---|---|---|---|---|---|---|
| | | Concentration (mg·kg$^{-1}$) | Sd | Cv | Concentration (mg·kg$^{-1}$) | Sd | Cv | Concentration (mg·kg$^{-1}$) | Sd | Cv |
| 1 | *Pueraria lobata* | 45.2±0.5 | 0.5 | 1.2% | 112.7±0.3 | 0.3 | 0.3% | 251.2±0.2 | 0.2 | 0.1% |
| 2 | *Lycium barbarum* | 80.9±0.3 | 0.3 | 0.4% | 137.5±0.2 | 0.2 | 0.2% | 204.8±0.5 | 0.5 | 0.2% |

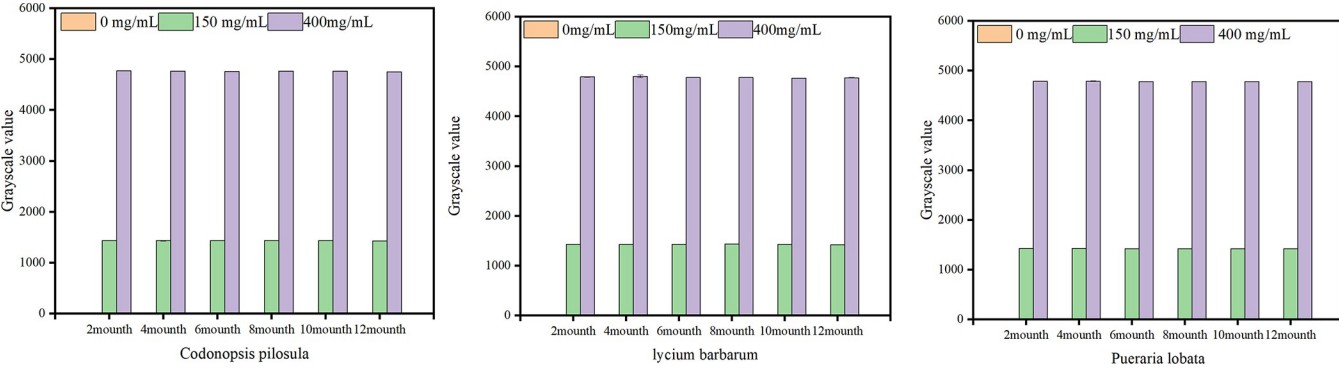

**Fig 5. The greyscale of sulphur dioxide and detection reagent storage time.** Three parallel samples were assayed for each targeted SO$_2$ concentration spike in each CHM, with a preservation stability trial conducted. Reagents for the detection process were maintained in a refrigerator set at 2–8˚C. Unopened reagents were retrieved at 0 months, 2 months, 4 months, 6 months, 8 months, 10 months, and 12 months post-storage initiation.

value for all three CHM concentrations remained consistent throughout the 12-month storage interval. Furthermore, this method was stable for up to one year and compatible with a wide range of CHM. The results are presented in Fig 5.

### 3.5 The resout of application verification

The results of 10 batches of Chinese herbal pieces according to this detection system were compared with those of ion chromatography, the third method of Chinese Pharmacopoeia (2015 edition). In one of the samples (*Lycium barbarum*), the concentration of SO$_2$ detected by this system was 896.5 mg/kg, which was significantly different from the concentration of 3073 mg/kg detected by the ion chromatography, this is because the concentration of sulfur dioxide residue exceeds the maximum detection limit of this detection system, but according to the judgment of the national standard, the results are also positive, and the detection results of ion chromatography are consistent, that is, its sulfur dioxide residue seriously exceeded the standard. Through the comparison of 10 different samples, the detection system and the ion chromatography detection results are consistent. The results are shown in Table 6.

## 4. Conclusions

It is important to note that this method relies on a colorimetric approach, which is contingent on the membrane's capacity to adsorb the colorant. Consequently, the detection range is

**Table 6. Detection results of uniformity for samples.**

| No | Name | Results/ The concentration of SO$_2$/(mg/kg) | |
| --- | --- | --- | --- |
| | | Lead acetate–based test strip | ion chromatography |
| 1 | *Lycium barbarum* | 896.5 | 3073 |
| 2 | *Angelica dahurica* | 777 | 782.2 |
| 3 | *Angelica sinensis* | 251.5 | 252 |
| 4 | *Safflower* | 12.05 | 11.6 |
| 5 | *Sha Yuanzi* | 8.05 | 7 |
| 6 | *Ganoderma lucidum* | 14.05 | 12.6 |
| 7 | *Epimedium* | 8.5 | 5.8 |
| 8 | *Basil leaves* | 6.45 | 4.6 |
| 9 | *Pueraria lobata* | 99.5 | 98.1 |
| 10 | *Codonopsis pilosula* | 165.5 | 164.7 |

somewhat constrained. Furthermore, the necessity to dilute the sample to ensure that the measured values fall within the detectable range can introduce dilution-related errors. In future work, we will choose a new support material to adsorb lead acetate to expand the detection range linearly. Although the application evaluation of this study shows that it is suitable for real sample detection, the application of this system in real scenes needs a lot of verification. The future development of this technology should focus on automation and high-throughput detection methods. We believe that these advancements would greatly improve the practicality and efficiency of this technique.

In this investigation, we established a rapid detection technique utilizing lead acetate test strips for the prompt identification of SO2 in traditional Chinese medicinal materials and herbal preparations. It can complete the accurate quantitative detection of sulfur dioxide residues in traditional Chinese medicine with only a few simple steps in a few tens of minutes, and can effectively eliminate the interference of the traditional Chinese medicine's color, and it is a stable, reliable, convenient and rapid method for the quantitative determination of sulfur dioxide in Chinese medicinal materials this developed technique may offer valuable guidance for the creation of similar test strips intended for quantitative detection purposes.

## Author Contributions

**Conceptualization:** Qiyu Wang, Cuixiang Xu.

**Formal analysis:** Penghua Zhao, Haixiang Zhang.

**Funding acquisition:** Penghua Zhao.

**Methodology:** Yaping Li.

**Resources:** Jianhua Wang.

**Validation:** Qiyu Wang.

**Visualization:** Dongliang Li.

**Writing – original draft:** Penghua Zhao.

**Writing – review & editing:** Penghua Zhao, Xiaoyan Huang.

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
