## [Decision Letter · Decision Letter 0]

9 Jul 2024

PONE-D-24-18964Lead acetate–based test strip method for rapid and quantitative detection of residual sulfur dioxide in Chinese herbal medicinesPLOS ONE

Dear Dr. Zhao,

Thank you for submitting your manuscript to PLOS ONE. After careful consideration, we feel that it has merit but does not fully meet PLOS ONE’s publication criteria as it currently stands. Therefore, we invite you to submit a revised version of the manuscript that addresses the points raised during the review process.

Thank you so much for your submission. Kindly prepare your revised paper through addressing all comments of the reviewers. I am looking forwarding to receiving your revised paper to re-evaluate for Plos One. ==============================

We look forward to receiving your revised manuscript.

Kind regards,

Tien Anh Tran

Academic Editor

PLOS ONE

2. PLOS requires an ORCID iD for the corresponding author in Editorial Manager on papers submitted after December 6th, 2016. Please ensure that you have an ORCID iD and that it is validated in Editorial Manager. To do this, go to ‘Update my Information’ (in the upper left-hand corner of the main menu), and click on the Fetch/Validate link next to the ORCID field. This will take you to the ORCID site and allow you to create a new iD or authenticate a pre-existing iD in Editorial Manager. Please see the following video for instructions on linking an ORCID iD to your Editorial Manager account: https://www.youtube.com/watch?v=_xcclfuvtxQ".

 [This work was supported by the Key R&D plan of Shaanxi Province (No.2021ZDLSF01-07), Shaanxi provincial People's Hospital Science and Technology Personnel Support Program(No.2021LJ-02), Key Research and development program of Shaanxi Province(No.2022KWZ-20), Shaanxi provincial People's Hospital Science and Technology Development Incubation Fund（2023YJ-22）Key Research and development program of Shaanxi Province（2024SF-YBXM-081）].  

"""

5. We note that you have indicated that there are restrictions to data sharing for this study. PLOS only allows data to be available upon request if there are legal or ethical restrictions on sharing data publicly. For more information on unacceptable data access restrictions, please see http://journals.plos.org/plosone/s/data-availability#loc-unacceptable-data-access-restrictions. 

5. We note that Figure 1 in your submission contain copyrighted images. All PLOS content is published under the Creative Commons Attribution License (CC BY 4.0), which means that the manuscript, images, and Supporting Information files will be freely available online, and any third party is permitted to access, download, copy, distribute, and use these materials in any way, even commercially, with proper attribution. For more information, see our copyright guidelines: http://journals.plos.org/plosone/s/licenses-and-copyright.

6. Please include your tables as part of your main manuscript and remove the individual files. Please note that supplementary tables (should remain/ be uploaded) as separate ""supporting information"" files".

Additional Editor Comments:

Thank you so much for your submission. After considering carefully the comments of the experts. Your paper must be improved and revised all comments of the experts. We are looking forward to receiving your revised manuscript.

Reviewers' comments:

Reviewer's Responses to Questions

**Comments to the Author**

1. Is the manuscript technically sound, and do the data support the conclusions?

Reviewer #1: Yes

Reviewer #2: Partly

Reviewer #3: Partly

Reviewer #4: Yes

2. Has the statistical analysis been performed appropriately and rigorously? 

Reviewer #1: No

Reviewer #2: Yes

Reviewer #3: N/A

Reviewer #4: Yes

3. Have the authors made all data underlying the findings in their manuscript fully available?

Reviewer #1: No

Reviewer #2: Yes

Reviewer #3: No

Reviewer #4: Yes

4. Is the manuscript presented in an intelligible fashion and written in standard English?

Reviewer #1: No

Reviewer #2: Yes

Reviewer #3: No

Reviewer #4: Yes

5. Review Comments to the Author

Reviewer #1: In this work, the authors introduce lead acetate–based test strip method for rapid and quantitative detection of residual sulfur dioxide in Chinese herbal medicines. The content is detailed and enriching, showcasing the authors' broad perspective. However, there are several issues within the article that require some modifications：

1. Please reorganize the abstract part. Background, Methods, Results and Conclusions should not be indicated.

2. Please rewrite the keyword ‘Quantitative’.

3. The section 2. Materials and methods need to be reorganized. There are too many subtitles which make experimental details unclear.

4. Please reorganize the results part by reducing subtitles and extract the main results followed by significant biological explication.

5. Please add scale in Figure 1.

6. Please add error bar in the experimental curves.

7. The figutes are not good presented.

8. The table 2 is not good enough.

In conclusion, the author shows enough expermental details and results to support the main idea. However, the work needs to be slightly improved. I recommend the publication after a major revision.

Reviewer #2: In section 3.2. Preparation of standard grayscale cards, teststrips with a lead acetate concentration of 15% were prepared. However, in section 3.1.2.1. and Figure 2a, lead acetate with concentration of 20% showed better linear pattern of the detected gray value.

In section 3.4.2.1. Precision testing, SO2 residual concentration for the three herbs was not clearly addressed in the method part.

The colors of the line on Figure 2a can’t correspond to the caption beside it.

Reviewer #3: In this manuscript, entitled “Lead acetate–based test strip method for rapid and quantitative detection of residual sulfur dioxide in Chinese herbal medicines”, the authors developed a rapid and quantitative sulfur dioxide detection technique for the evaluation of sulfur dioxide residues in Chinese herbal medicines. The authors converted the sulfur dioxide to hydrogen sulfide, which was then detected using lead acetate test paper. My questions and suggestions are as follows:

1. In “Introduction” section:

Why don't you refer to the latest edition of Chinese Pharmacopoeia (2020)?

In addition to the mentioned common detection methods in addition to acid-base titration, ion chromatography, gas chromatography, fluorescence derivatization, colorimetry, etc., there are also many methods for the rapid detection of sulfur dioxide residue in Chinese herbal medicine. Please provide an overview of the latest research to highlight the need for the detection methods in this study.

2. In “Methods” section:

Please attach the chemical reaction formula.

What was the reason for choosing sodium borohydride (NaBH₄) as the reducing agent? The reaction between sulfur dioxide or sulfite and sodium borohydride is not a common chemical reaction because their chemical properties do not directly promote such reactions under normal conditions. However, under certain conditions, the two compounds may react, but this is rare and the reaction conditions are usually harsh. What reaction conditions were used in this study to facilitate the reaction? This point is not seen in the article at present!

Why not use real Chinese medicine for research? In this study, what is the role of Chinese herbal medicine extract in it? Is it just a simulation? The real detection method will carry out pre-extraction treatment of Chinese medicinal materials, and the pre-treatment steps will also affect the reaction conditions. Therefore, the use of Chinese medicinal materials for real detection is an essential part of the establishment of the new methodology!

It is suggested that a batch of Chinese medicinal materials should be simultaneously tested by other known mature methods, such as acid-base titration in Chinese Pharmacopoeia, in order to demonstrate the accuracy of the detection results of this method.

3. In “3.1.1”:

In this part, the best reaction time was determined to be 10 min. Why was 15 min used in 3.1.2.1. to detect the influence of different concentrations of lead acetate on the detection accuracy?

4. In “3.1.2.1”:

In this part, CN95 membrane was used to detect the influence of different concentrations of lead acetate on the detection accuracy. However, the best experimental membrane type was determined as CN 140 in 3.1.2.2. Please explain the different membrane types of the two parts!

Should the optimal membrane type be determined first, and then the influence of the concentration of lead acetate under the optimal membrane type on the detection results be tested?

5. In “Discussion” section:

There is no discussion section in this paper. Please supplement the latest progress of relevant technology in the current research field, compare it with this study, and discuss the limitations and future research direction of this technology!

Reviewer #4: The paper has established a rapid detection technique utilizing lead acetate test strips for the prompt identification of SO2 in traditional Chinese medicinal materials and herbal preparations. The idea is interesting and data is fine. I Just have two questions.

1. Why the membrane type has effect on detection accuracy? The authors should address it in the context.

2. The language need to be improved and polished.

6. PLOS authors have the option to publish the peer review history of their article (what does this mean?). If published, this will include your full peer review and any attached files.

Reviewer #1: No

Reviewer #2: **Yes: **HSU, SHUO-MIN

Reviewer #3: No

Reviewer #4: No

---

## [Author Response · Author response to Decision Letter 0]

22 Aug 2024

I ensure that my manuscript meets PLOS ONE's style requirements, including those for file naming.

I ensure that I have an ORCID iD and that it is validated in Editorial Manager. 

All of the authors approved the submitted version (and any substantially modified version that involves the author’s contribution to the study) and All of the authors agrees to be personally accountable for the author’s contributions AND to ensure that questions related to the accuracy or integrity of any part of the work are appropriately investigated, resolved, and the resolution documented in the literature

This work was supported by the Key Research and development program of Shaanxi Province(2024SF-YBXM-081), Shaanxi provincial People's Hospital Science and Technology Personnel Support Program(No.2021LJ-02), Key Research and development program of Shaanxi Province(No.2022KWZ-20), Shaanxi provincial People's Hospital Science and Technology Development Incubation Fund（2023YJ-22）Shaanxi Provincial health and Medical Research Innovation Capacity Enhancement Plan(2024PT-01), Shaanxi Province Innovation Capability Support Program Project (S2024-ZC-TD-0127), Shaanxi Provincial health high-level talent cultivation program, the Shaanxi Special Support Plan for High Level Talents. The funders had no role in study design, data collection and analysis, decision to publish, or preparation of the manuscript.

We did not multiple submissions for one draft, this is our original article. We corrected all the comments above point to point. See the corrected article for details

The Figure 1 in our submission contain copyrighted images. Thank you for your insightful comments. We appreciate your guidance and understand the importance of adhering to the CC BY4.0 license requirements. The figure in this paper is similar but not identical to the original image for illustrative purposes only, we have clearly indicated this in the figure caption, including the source information as required.

We assure you that we will address these matters with the utmost diligence and professionalism. Thank you for your understanding and support throughout the peer-review process.

We revised on the reviewer's suggestion regarding the inclusion of tables within the main text of our manuscript and the retention of supplementary tables as separate "supporting information" files.

---

## [Decision Letter · Decision Letter 1]

4 Sep 2024

Lead acetate–based test strip method for rapid and quantitative detection of residual sulfur dioxide in Chinese herbal medicines

PONE-D-24-18964R1

Dear Dr. Zhao,

We’re pleased to inform you that your manuscript has been judged scientifically suitable for publication and will be formally accepted for publication once it meets all outstanding technical requirements.

Kind regards,

Tien Anh Tran

Academic Editor

PLOS ONE

Additional Editor Comments (optional):

After reviewing carefully your revised paper and the recommendations of reviewers, the current manuscript could be accepted to publish on Plos One.

Reviewers' comments:

Reviewer's Responses to Questions

**Comments to the Author**

1. If the authors have adequately addressed your comments raised in a previous round of review and you feel that this manuscript is now acceptable for publication, you may indicate that here to bypass the “Comments to the Author” section, enter your conflict of interest statement in the “Confidential to Editor” section, and submit your "Accept" recommendation.

Reviewer #2: All comments have been addressed

Reviewer #4: All comments have been addressed

2. Is the manuscript technically sound, and do the data support the conclusions?

Reviewer #2: Yes

Reviewer #4: Yes

3. Has the statistical analysis been performed appropriately and rigorously? 

Reviewer #2: Yes

Reviewer #4: Yes

4. Have the authors made all data underlying the findings in their manuscript fully available?

Reviewer #2: Yes

Reviewer #4: Yes

5. Is the manuscript presented in an intelligible fashion and written in standard English?

Reviewer #2: Yes

Reviewer #4: Yes

6. Review Comments to the Author

Reviewer #2: (No Response)

Reviewer #4: As for the revision, I think all my questions are answered and I think the manuscript is acceptable now.

7. PLOS authors have the option to publish the peer review history of their article (what does this mean?). If published, this will include your full peer review and any attached files.

Reviewer #2: No

Reviewer #4: No

---

## [Editor Report · Acceptance letter]

31 Oct 2024

PONE-D-24-18964R1 

PLOS ONE

Dear Dr. Zhao, 

I'm pleased to inform you that your manuscript has been deemed suitable for publication in PLOS ONE. Congratulations! Your manuscript is now being handed over to our production team.

Kind regards, 

on behalf of

Professor Tien Anh Tran 

Academic Editor

PLOS ONE